# Animal Model for Lower Urinary Tract Dysfunction in Parkinson’s Disease

**DOI:** 10.3390/ijms21186520

**Published:** 2020-09-07

**Authors:** Takeya Kitta, Mifuka Ouchi, Hiroki Chiba, Madoka Higuchi, Mio Togo, Yui Abe-Takahashi, Naohisa Kusakabe, Nobuo Shinohara

**Affiliations:** 1Department of Renal and Genitourinary Surgery, Graduate School of Medicine, Hokkaido University, Sapporo, Hokkaido 060-8648, Japan; hirokic@mvf.biglobe.ne.jp (H.C.); mado-chanchan@kxa.biglobe.ne.jp (M.H.); mio.t.g.89@gmail.com (M.T.); kussanaochan66@yahoo.co.jp (N.K.); nozomis@mbj.nifty.com (N.S.); 2School of Rehabilitation Sciences, Health Sciences University of Hokkaido, Tobetsu 061-0293, Japan; m-ouchi@hoku-iryo-u.ac.jp; 3Department of Physical Therapy, Faculty of Health Sciences Hokkaido University of Science, Sapporo 006-8585, Japan; abe-y@hus.ac.jp

**Keywords:** Parkinson’s disease, animal model, lower urinary tract dysfunction, non-motor symptom, overactive bladder, detrusor overactivity

## Abstract

Although Parkinson’s disease (PD) is characterized by the loss of dopaminergic neurons in the substantia nigra and subsequent motor symptoms, various non-motor symptoms often precede these other symptoms. While motor symptoms are certainly burdensome, a wide range of non-motor symptoms have emerged as the key determinant of the quality of life in PD patients. The prevalence of lower urinary tract symptoms differs according to the study, with ranges between 27% and 63.9%. These can be influenced by the stage of disease, the presence of lower urinary tract-related comorbidities, and parallels with other manifestations of autonomic dysfunction. Animal models can provide a platform for investigating the mechanisms of PD-related dysfunction and for the assessment of novel treatment strategies. Animal research efforts have been primarily focused on PD motor signs and symptoms. However, the etiology of lower urinary tract dysfunction in PD has yet to be definitively clarified. Several animal PD models are available, each of which has a different effect on the autonomic nervous system. In this article, we review the various lower urinary tract dysfunction animal PD models. We additionally discuss techniques for determining the appropriate model for evaluating the development of lower urinary tract dysfunction treatments.

## 1. Introduction

Parkinson’s disease (PD) is characterized by the loss of dopaminergic neurons in the substantia nigra that is often followed by the appearance of subsequent motor and various non-motor symptoms. Recent research has focused more attention on non-motor symptoms as compared to previous research, with a systematic review undertaken in 2006 [1]. Patients with PD often have lower urinary tract symptoms (LUTS), such as increased urinary frequency, urinary incontinence and nocturia, which overlap with those of the overactive bladder symptoms. Previous studies have reported the incidence of these symptoms ranges from 27% to 63.9% [2]. LUTS also have a significant negative effect on the quality of life [3,4]. Animal and clinical research efforts have largely focused on PD motor signs and symptoms, as they are the basis for PD diagnosis. However, the etiology of lower urinary tract dysfunction in PD has yet to be fully clarified. As a result, various animal studies have recently begun to investigate lower urinary tract dysfunction.

Recently, there have been many notable changes in PD treatments. Commonly prescribed medications for PD include levodopa and various dopaminergic agonists. In some but not all patients, bladder overactivity can be improved through the use of dopamine replacement therapy. The effects of PD treatments are complex and perhaps biphasic. It has also been postulated that dopaminergic D1 and D2 receptors, auto-receptors and post-synaptic sites and both central and peripheral dopamine receptor stimulation may be involved in these effects [5] (a more detailed explanation of dopaminergic receptors is provided later in this review.) Prior to starting clinical trials, the effects and toxicities of developed drugs are usually confirmed through the use of animal experiments. Thus, animal models are of crucial importance in gaining a better understanding of lower urinary tract dysfunction and for testing the efficacy and safety of promising treatment options.

In this paper, we discuss the current literature on animal models of PD-associated lower urinary tract dysfunction.

## 2. LUTS and Animal Models

Overactive bladder (OAB) symptoms, such as urinary urgency, frequency and nocturia, with or without urgency incontinence, are the most common LUTS in PD patients [6]. The most common finding of urodynamic studies (bladder function study) in PD patients is detrusor overactivity (DO). This is shown by uninhibited contractions that occur during the storage phase (bladder filling), which is the result of impaired activity of the central nervous system, especially in the nigrostriatal dopaminergic pathways [7]. Previous animal studies using a 6-hydroxydopamine (6-OHDA)-induced PD model have been able to demonstrate the developed DO-like cytometric reaction [8]. Furthermore, detrusor sphincter dyssynergia (DSD) is a huge problem for treating the lower urinary tract dysfunction in the patient with PD. In recent reports, DSD is a much less common condition in Parkinson’s disease. DSD was observed on voiding at a rate of 0–3%.

### OAB and DO

Lower urinary tract dysfunction terminology has yet to be standardized in animal studies. In addition, it has yet to be clarified which of the urodynamic findings in animals directly correspond to human lower urinary tract dysfunction conditions. Based on the International Continence Society (ICS) standardization, the term DO can only be used during cystometry when animals exhibit non-voiding bladder contractions during the storage phase. In other cases in other voiding behavior studies when frequent voiding with reduced voided volume or reduced intercontraction intervals with decreased bladder capacity is observed during cystometry, this condition is referred to as “bladder overactivity” (a not overactive bladder) in order to avoid confusion with DO. OAB is a clinical diagnosis based on the presence of urinary urgency as the key symptom. In a human study, OAB has been shown to be highly prevalent and overlap with DO. However, DO and OAB are not synonymous, as a cystometry study determined that as many as 50–75% of men with benign prostatic hyperplasia have concomitant DO [9]. In fact, in humans with normal lower urinary tracts, DO was observed during urodynamic studies. Even so, urgency cannot be ascertained or quantified in animals, even if there are voiding behavior changes. As a result, it is widely recognized that surrogate markers need to be used during these situations. Thus, this is the reason why most experimental models are used as non-voiding bladder contractions for OAB symptoms in animals [10].

## 3. Methods for Assessing Lower Urinary Tract Function

In the past, lower urinary tract function has been investigated in a variety of animal models, although primarily using cats [11], with more recent studies mainly focusing on rodents due to the advantages of well-defined genetic lineages, straightforward gene manipulation, short lifespan and minimal holding space. All of these factors help to increase the quality and reduce the overall costs of these experiments. Thus, lower urinary tract function in animals can be assessed using a variety of behavioral experiments. 

### 3.1. Cystometry

The use of cystometry during urodynamic testing is a traditional method for detecting changes in the bladder function in animals [12]. Rats and mice are two of the primary species that have been used to evaluate urodynamic parameters. However, rats are the most widely used animals for such experiments due to the relative ease of use for the technology involved compared to that for mice [13]. Even so, the mouse bladder is more comparable to the human bladder, as they have intramural ganglia in the bladder. Unfortunately, performing cystometry in mice can be technically difficult due to the small bladder capacity (about 0.1 mL, even in adult mice) [13]. The development of genetic modification techniques has led to an increasing use of knockout (KO) models and other transgenic mice in the establishment of experimental models of LUTS.

Bladder function is evaluated using the following parameters: voided volume, post-voiding residual urine volume (the bladder is emptied with suction via a syringe after micturition), bladder capacity (voided volume + residual volume), voiding efficiency (voided volume/bladder capacity × 100), micturition pressure (maximum voiding pressure), the number of non-voiding contractions and threshold pressure (bladder pressure at the start of detrusor contraction for micturition) (Figure 1).

### 3.2. Urination Patterns on Filter Paper

Although cystometry is one of the most helpful procedures for measuring urinary bladder function, it cannot replicate physiological conditions. Furthermore, anesthesia may additionally affect bladder capacity and diurnal behavior. To avoid these potential issues, some researchers have assessed urination patterns on filter paper, which does not require the use of an intravesical catheter [14,15]. In these studies, after mice are habituated to their cages, urination patterns are evaluated by examining the spots that form on filter paper, with these urine spots photographed under UV light [16]. This detailed procedure was devised as a way to record the micturition of a mouse that had free movement within a cage, including access to food and water during the experiment days [17]. Furthermore, a system using an automated voided stain on paper was developed in order to mimic the frequency-volume chart that is used in a human clinic, which has proven to be an excellent translational procedure. In this system, a rolled laminated filter paper, pre-treated to turn the edge of urine stains deep purple, is wound up at a speed of 10 cm per hour under a water-repellent wire lattice. Urine stains are then counted and traced in order to convert micturition volume according to the formula of a standard curve, which is calculated by the correlation of normal saline and the stained area ranging from 10 to 800 μL [18].

### 3.3. Urethral Function

To date, almost all relevant studies have primarily focused on bladder functions and not on urethral functions. However, this is an important topic, as continence mechanisms involve both the bladder and the urethra [19]. Furthermore, DSD developed in animals with a complete suprasacral spinal cord injury model. However, for now, we cannot use a DSD model in the animal mode of PD.

#### 3.3.1. Urethral Pressure Amplitude during Electrical Stimulation and Urethral Baseline Pressure (UBP)

UBP and the urethral pressure amplitude during electrical stimulation (A-URE) are recorded. UBP is defined as the flat section of the pressure recording that is seen just before the response to the electrical stimulation. The A-URE is defined as the average value of the maximal urethral pressure change from the UBP when electrical stimulations are administered in order to cause pressure changes from the UBP (Figure 2).

#### 3.3.2. Leak Point Pressure

The leak point pressure is measured after slowly increasing the abdominal compression pressure while in the supine position. The pressure is applied to the abdominal wall until fluid begins to leak from the urethral orifice, at which time the compression is stopped. Peak pressure is measured via an intravesical catheter.

## 4. PD Models 

PD is a multifactorial disease caused by the sum of the impacts of multiple environmental and genetic factors. These multiple environmental factors include inflammation, viruses and toxicants. For example, the pesticide rotenone is a mitochondrial toxin [20,21]. At the present time, PD experimental models can be roughly classified into two categories: toxin-based and genetic models (Table 1). In this paper, we first explain the mechanisms of the PD models, and then discuss the results of the lower urinary tract function obtained from each of these animal models.

### 4.1. Toxin-Based Models (Classic PD Models)

Toxin-based models induce the rapid degeneration of the nigrostriatal dopaminergic neurons, which mimics sporadic PD. 1-Methyl-4-phenyl-1,2,3,6-tetrahydropyridine (MPTP) and 6-hydroxydopamine (6-OHDA) also act as neurotoxins against dopamine neurons [34,35]. Toxin-based animal models have greatly contributed to the development of symptomatic treatments, primarily for the evaluation of motor symptoms. In addition, since the phenotype is clear and the experimental time is short, these toxins can be applied to various animals.

Since MPTP is a lipophilic molecule, it can easily cross the blood–brain barrier. After systemic administration, monoamine oxidase B in astrocytes can oxidize MPTP into the potent dopaminergic neurotoxin 1-methyl-4-phenylpyridinium ion. Due to the structural similarity of this substance to dopamine, it is a toxic metabolite that is readily absorbable by the dopaminergic neuron through the dopamine transporter [36].

One of the most commonly used PD models is the 6-OHDA animal model, which was established in the late 1960s by Urban Ungerstedt [37]. This model is based on the dopamine analog and neurotoxin 6-OHDA, which is injected into the substantia nigra (usually unilateral injection), the medial forebrain bundle or the striatum, in order to create a substantial dopamine loss in the brain similar to that observed in PD patients [38]. However, both the MPTP and 6-OHDA models lack the neuropathological feature of PD, which is the formation of Lewy Bodies (LB) [39].

### 4.2. Genetic Models

One of the most important genetic factors in PD is SNCA (α-synuclein), which was the first gene identified in familial PD [40]. Mutations of SNCA have been characterized in the inherited form of PD, for example, substitutions (A53T, A30P and E46K), duplication or triplication [41]. SNCA has also been importantly identified as the major building block of LB. Based on these significant findings, researchers have started to model PD based on the overexpression of the wild type or mutant forms of SNCA in animals. Many α-synuclein transgenic animal models have been proposed based on this model, with the subsequent experimental results revealing the pathology of PD [42,43]. The SNCA transgenic model can reproduce SNCA aggregation similar to that found in human PD. However, this model usually requires a long time and in addition, it is sometimes difficult to obtain the desirable phenotypes compared to those for the toxin-based model. Following the detection of several familial PD-linked genes (LRRK2 [44], DJ-1 [45] and PINK1 [46]), this led to the use of LRRK2 models [47], DJ-1 KO mice [48] and PINK1 KO mouse models [49].

## 5. Non-Human Primates (NHP)

In medical research, NHPs have played a critical role in providing significant insights into the mechanism of disease, as NHPs are closely related to humans genetically and exhibit symptoms similar to those found in humans [50]. However, studies of NHPs require high resource support and expertise, in addition to being time consuming. To date, only an estimated 10% of PD research is being carried out in NHPs. Due to both the high cost and ethical issues, NHP studies are often performed for preclinical evaluation of therapies [51]. MPTP injected in marmosets [52] or in cynomolgus macaques [24] similarly provoked PD-like motor symptoms accompanied by bladder overactivity shown by frequent urination with reduced voided volume [23]. Based on their positive or negative coupling to adenylyl cyclase, central dopamine receptors are categorized into two subfamilies, the D1-like and D2-like dopamine receptors (hereafter simply called D1 and D2). However, five subtypes of dopamine receptors have been identified in various dopaminergic systems of mammalian brains [53]. Dopamine receptor agonists acting on both dopamine D1 and D2 receptors have been demonstrated to usually have therapeutic effects in an NHP model of MPTP-induced parkinsonism. However, the effects of dopamine receptor agonists on the lower urinary tract function are complex. Yoshimura et al. reported that in MPTP-treated monkeys [23], a subcutaneous injection of SKF 38393, which is a dopamine D1 receptor agonist, significantly increased the bladder volume and pressure thresholds for inducing the micturition reflex without affecting the reflex in normal monkeys. In contrast, subcutaneous injections of quinpirole, which is a dopamine D2 receptor agonist, and apomorphine, which is a dopamine D1 and D2 receptor agonist, slightly, but significantly reduced the volume threshold of the bladder for the micturition reflex in both normal and MPTP-treated groups. Thus, it seems likely that both dopamine D1 and D2 receptors mediate opposite effects on the micturition reflex.

## 6. Rodents

Rodents, such as rats and mice, make up 90% of all laboratory animals. These animals are biologically similar to humans and have a shortened generation period and easy-to-control environmental factors, which can influence experimental studies [54]. As a result, many researchers in the PD field prefer the use of rodent animal models.

### 6.1. Rats

A rat PD model induced by a unilateral injection of 6-OHDA into the substantia nigra results in bladder overactivity [35,55]. Furthermore, a 6-OHDA induced lesion in the medial forebrain bundle was also shown to cause bladder dysfunction with an early onset (starting from 3 days after the 6-OHDA injection) [27]. Seki et al. reported that the dopaminergic mechanisms involved in lower urinary tract dysfunction could additionally include a failure to activate D1 receptors and/or an overactivation of D2 receptors [5]. Other studies using these animal models have shown that bladder overactivity was suppressed by enhancement of D1 receptors with SKF 38393 or pergolide [23,55], which suggests that bladder overactivity in PD is primarily induced by disruption of D1 dopamine receptor-mediated inhibition of the micturition reflex. These reports match the results for a previous study of performed in NHP [23]. The dopaminergic mechanisms involved in bladder control include the D1 receptor activity that inhibits the micturition reflex and the D2 receptor activity that facilitates the micturition reflex in both PD and in normal rats [7,8]. Determining the effect in the dopamine receptor study is not simple, since it is associated with not only the site of the dopaminergic receptor, auto-receptors and post-synaptic sites but also the central and peripheral dopamine receptor stimulation. To clarify this complex mechanism, Yoshimura et al. compared the effect of each of the dopaminergic agonist/antagonist administration routes (Table 2) [5,8].

Treatment of PD patients using levodopa can sometimes cause a worsening of LUTS due to the activation of dopamine D2 receptors [56]. In addition, bladder overactivity was shown to be suppressed by the adenosine A2A receptor antagonist, ZM 241385, in a rat model, which suggested that enhanced activity of the adenosine A2A system in the brain contributed to bladder overactivity associated with PD [30]. It is assumed that the adenosine A2A receptor-expressing neural pathways are located downstream of the D1 receptor expressing pathways involved in the control of micturition. This is because results from a rat PD model have demonstrated that inhibition of bladder activity by D1 receptor activation was able to induce partial suppression of the adenosine A2A receptor-mediated excitatory mechanisms [30]. Based on these findings, our group proposed that the bladder overactivity associated with PD is at least in part induced by the enhanced activity of the adenosine A2A receptors in the brain, while the dopamine D1 receptor mediated inhibition of bladder overactivity involves the suppression of the A2A receptor activation in PD patients (Figure 3).

Our group also reported that intravenous ZM 241385 dose-dependently increased the amplitude of evoked potentials in the anterior cingulate cortex in a rat PD model but not in sham operated rats. This suggests that the anterior cingulate cortex neurons have an inhibitory role in bladder control in addition to an executive function, which includes decision-making in the micturition reflex [26]. Clinical studies of adenosine A2A receptor antagonists have also provided promising results for the treatment of motor dysfunction in PD subjects [57,58]. Istradefylline, which is an adenosine A2A receptor antagonist, has already been approved and launched for patients with PD. Furthermore, this may also be a promising candidate for treatment of lower urinary tract dysfunction in PD patients [59]. In support of this assumption, a recent open-label clinical study reported that treatment with istradefylline for 12 weeks significantly improved LUTS, such as nocturia and urgency in 13 male PD patients [59]. However, a larger-sized, placebo-controlled randomized study will need to be undertaken in order to confirm these results. Furthermore, our group also recently reported that significant improvements were observed during a 1-year istradefylline treatment of LUTS in PD patients [60]. Moreover, istradefylline may not only improve motor symptoms, but also LUTS in PD patients.

Although urinary continence mechanisms involve the bladder and urethra coordination, urethral function is rarely discussed by itself. Clinically, both urge urinary incontinence (OAB) and stress urinary incontinence are reported in PD patients [61], with the latter occurring when the bladder pressure exceeds the urethral pressure at the moment of an increased abdominal pressure. Ouchi et al. reported finding that active urethral closure was significantly impaired in PD rats, a finding that may provide insights into the underlying mechanisms of urinary incontinence in PD patients [19] (Figure 4). In addition, these intriguing findings suggest that activated dopamine D1 receptors in the central nervous system may have a compensating functional role that opposes urinary incontinence when insufficient dopamine is present, whereas activation of the dopamine D2 receptors may exacerbate urinary leakage due to a decline in the urethral pressure in the presence of increasing intra-abdominal pressure [19].

It has been suggested that not only is there an alteration of the central dopamine control of the micturition reflex in PD, but also there is a local contractile function of the urinary bladder [28]. Mitra et al. reported finding there was an altered contractile function caused by local morphological changes and plasticity in the urinary bladder in a rat PD model. Advances in stem cell research have raised hopes of establishing novel cell replacement therapies for PD. Furthermore, other reports have shown that bladder overactivity in 6-OHDA rats was attenuated by stem cell transplantations, with the transplantation of stem cells into the medial forebrain bundle leading to improvement of cystometric parameters during bladder dysfunction [29,62].

### 6.2. Mouse

In the α-synuclein overexpressing transgenic mice, the predominant bladder dysfunctions observed included urinary bladder hyperreflexia with increased voiding frequency, decreased voided volumes and the presence of non-voiding contractions [14]. Moreover, it was also confirmed that α-synuclein transgenic mice exhibited an early onset of lower urinary tract dysfunction. GM2 synthase KO mice also have both motor and non-motor PD symptoms. These GM2 KO mice have been shown to have a greater number of void spots compared to wild type mice, especially for mice at younger ages. Vidal-Martinez et al. confirmed the improvement of bladder function after the administration of FTY720, which is a novel immunomodulator that is able to block T-cell egress from lymph nodes, in GM2 +/− mice [33]. However, compared to the rat model studies, there have been fewer studies of PD completed in mice. Thus, in future, the use of KO mice will be crucial to achieve a better understanding of lower urinary tract dysfunction and in exploring the efficacy and safety of promising treatment options.

## 7. Limitations

We reviewed animal models for lower urinary tract dysfunction in PD. However, to date, no ideal animal model has been agreed upon for PD research. Furthermore, it is difficult to develop a model that can fully recapitulate the features of human PD, which usually takes years to manifest. Pre-clinical studies are vital to ensure safety and efficacy of new treatments before they are widely used in clinical practice; however, there are significant differences in the anatomy and biomechanics of different animal models and humans. Obviously, animal study must be refined or altered in any way possible so as to decrease potential for suffering for all involved animals. Despite these limitations, the current animal models can provide a useful platform for selectively studying the pathophysiology and the interactions of the multiple etiologic factors involved in PD.

## 8. Conclusions

Various toxin-based and genetic animal models have been created for the purpose of conducting PD studies. However, each model has both advantages and disadvantages with regard to exploring the mechanism of lower urinary tract function. Based on the dopamine depletion model, results have shown that D1 receptor activity inhibits the micturition reflex, while D2 receptor activity facilitates the micturition reflex in the PD model. Thus, there is a complex mechanism that is responsible for controlling lower urinary tract dysfunction when using levodopa and other various dopaminergic agonists for the treatment of motor symptoms in the real world. Furthermore, new therapies, such as deep brain stimulation and stem cell therapy, could additionally have effects on the lower urinary tract function. Therefore, the use of animal model studies is essential for understanding lower urinary tract dysfunction and for testing the efficacy and safety of promising treatment options.

## 9. Take Home Message

We reviewed various types of animal model for lower urinary tract dysfunction in PD. Animal research efforts have been primarily focused on PD motor signs and symptoms. However, almost all models also have non-motor dysfunction in these models. To detect lower urinary tract dysfunction, many researchers focused on both bladder function and urethral function, and refined procedures were reported. These techniques were established and comparable to clinical human study. At the present time, PD experimental models can be roughly classified into two categories: toxin-based and genetic models. All have their distinctive characteristics, so we have to fully understand each model. Furthermore, many researchers in the PD field prefer the use of rodent animal models, before confirming concept and treatment effectiveness in a large animal study. Recently, some groups revealed new treatment option for lower urinary tract dysfunction in PD using animal models. The use of animal model studies is essential for understanding lower urinary tract dysfunction and for testing the efficacy and safety of promising treatment options.

## Figures and Tables

**Figure 1 ijms-21-06520-f001:**
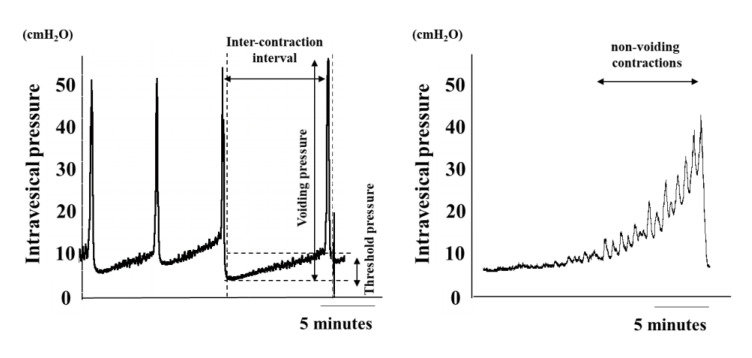
Cystometry parameters. Typical cystometric chart in rats. [10].

**Figure 2 ijms-21-06520-f002:**
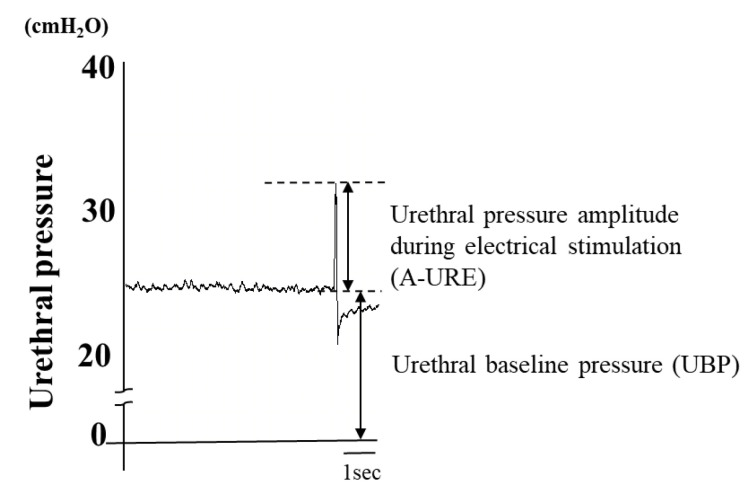
Urethral pressure amplitude during electrical stimulation (A-URE) and urethral baseline pressure (UBP). The urethral baseline pressure (UBP) was defined as the flat section of the pressure recording that is seen just before the response to the electrical stimulation. The amplitude during the electrical stimulation (A-URE) was defined as the average value of the maximal urethral pressure change from the UBP when electrical stimulations were given to cause pressure changes from the UBP [19].

**Figure 3 ijms-21-06520-f003:**
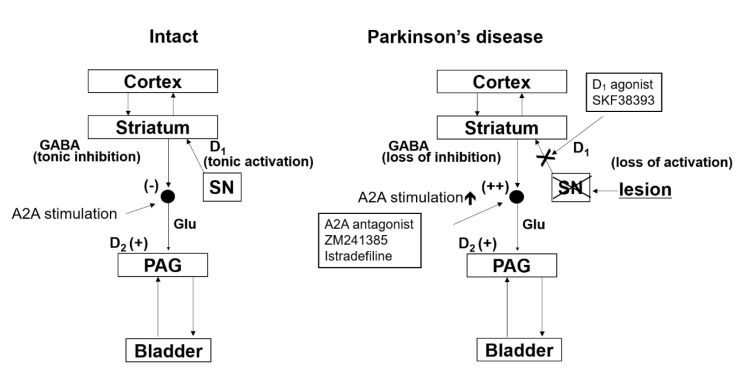
Working model of bladder dysfunction in PD. A hypothetical diagram demonstrates a working model of bladder dysfunction in PD. Micturition reflex is controlled by spinobulbospinal pathways through PAG in the midbrain and PMC in the brainstem. This neural circuit is under the control of higher centers, including the anterior cingulate cortex (ACC) and other cortex regions. (Intact) Under normal conditions, tonic inhibition from ACC suppresses the micturition reflex. Tonic firing (+) of dopaminergic neurons in the substantia nigra pars compacta (SN) activates the dopamine D1 receptors expressed on GABAergic inhibitory neurons in the striatum to induce tonic GABAergic inhibition (−) of the micturition reflex at the level of PAG. At the same time, D1 receptor stimulation suppresses the activity of adenosinergic neurons, which exert an excitatory effect on micturition via adenosine A2A receptors (+). (Parkinson‘s disease) In PD, dopaminergic neurons in the SN are lost (lesion), leading to the loss of dopamine D1 receptors activation (D1 (loss of activation)), which results in reduced activation inhibitory GABAergic neurons in the striatum (GABA (loss of inhibition)). At the same time, reduced D1 receptor stimulation enhances the adenosinergic mechanism to stimulate adenosine A2A receptors (A2A stimulation (++)), leading to facilitation of the spinobulbospinal pathway controlling the micturition reflex pathway. Administration of dopamine D1 receptor agonist (SKF 38393) can restore the GABAergic nerve activity and suppress A2A receptor-mediated activation to reduce bladder overactivity in PD. Furthermore, administration of adenosine A2A antagonists (ZM241385 or istradefylline) can suppress A2A receptor-mediated activation of the micturition reflex to reduce bladder overactivity in PD. Dopamine D2 receptors (D2 (+)) expressed in the spinal cord enhances the micturition reflex.

**Figure 4 ijms-21-06520-f004:**
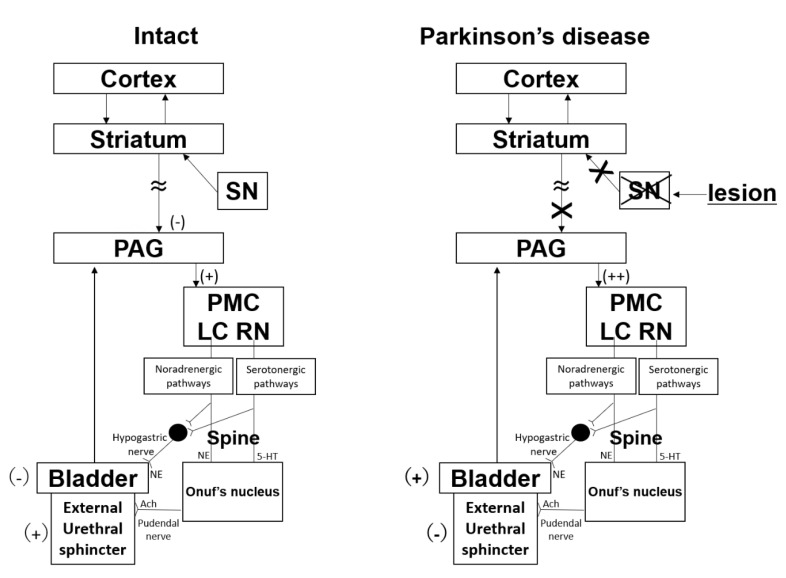
Working model for urethral dysfunction in PD. Schematic of our working model for urethral dysfunction. The active urethral closure mechanisms are regulated by the spinobulbospinal pathway. Intact: Descending signals of serotonergic and noradrenergic pathways from neurons in the raphe nucleus (RN) and locus coeruleus (LC), respectively. This neural circuit is under the control of the supraspinal regions and maintains urethral closure mechanisms. The descending signals travel downward to Onuf’s nucleus to innervate the external urethral sphincter and the pelvic floor muscle. Therefore, this neural system maintains active urethral closure mechanisms in the presence of increased inter-abdominal pressure. Under intact conditions, tonic inhibition (–) from supra-mesaticephalic sites suppresses bladder overactivity (+) and enhances urethral pressure (–). Parkinson‘s disease: PD dopaminergic neurons in the substantia nigra pars compacta (SN) are lost, and excitation (+) from the striatum is, therefore, decreased, leading to loss of the inhibitory functions of supra-mesaticephalic sites, including those of the cortex and supplementary motor areas (SMA). The loss of control results in facilitation of bladder overactivity (+) and inhibition of active urethral responses against increased intraabdominal pressure (–). 5-HT, 5-hydroxytryptamine; NE, norepinephrine; PAG, periaqueductal grey; PMC, pontine micturition center.

**Table 1 ijms-21-06520-t001:** Summary of animal models, assessment of the lower urinary tract function methods, intervention and results of Parkinson’s Disease (PD) research reports.

**Toxin-Based Models**					
**Model**	**Animal**	**Assessing Method of Lower Urinary Tract Function**	**Intervention**	**Results**	**Ref**
MPTP	marmoset	In vitro organ bath experiments (bladder)		enhanced spontaneous contractile activity and contractility in response to electrical field stimulation	[22]
MPTP	monkey	cystometry	dopamine D1 receptor agonist	increased the bladder volume and pressure thresholds for inducing the micturition reflex	[23]
			dopamine D2 receptor agonist	reduced the volume threshold of the bladder for the micturition reflex	
MPTP	monkey	cystometry	nonselective D1/D2 receptor agonist (pergolide)	reduced the bladder volume threshold, but increased the volume threshold	[24]
			D1/D2 agonist (5R,8R,10R)-6-methyl-8-(1,2,4-triazol-1-ylmethyl) ergoline maleate (BAM-1110)	increased the bladder volume threshold without significant effects on the micturition reflex	
MPTP	minipig	cystometry	deep brain stimulation	changed bladder capacity and compliance	[25]
6-OHDA (substantia nigra)	rat	cystometry	adenosine A2A receptor antagonist (ZM241385)	Neural activity in the anterior cingulate cortex was significantly increased along with suppression of bladder overactivity	[26]
6-OHDA (medial forebrain bundle)	rat	cystometry		injection of 6-OHDA into the medial forebrain bundle of rats causes development of bladder dysfunction	[27]
6-OHDA (medial forebrain bundle)	rat	In vitro organ bath experiments (bladder)		contractile response following electrical field stimulation was significantly higher	[28]
6-OHDA (medial forebrain bundle)	rat	cystometry	stem cell transplantation	urodynamical improve (lower threshold and intermicturition pressure, spontaneous activity and AUC*1)	[29]
6-OHDA (medial forebrain bundle)	rat	cystometry	stem cell transplantation	temporarily ameliorated bladder dysfunction	[30]
6-OHDA (substantia nigra)	rat	cystometry	dopamine D1 receptor agonist	increased bladder capacity	[8]
			dopamine D2 receptor agonist	reduced bladder capacity	
6-OHDA (substantia nigra)	rat	cystometry	dopamine D1 receptor agonist, D1 receptor antagonist, D2 receptor agonist and D2 receptor antagonist	dopaminergic modulation mediated by D1 receptors in the periaqueductal gray is responsible for the micturition reflex, in which a GABAergic mechanism is involved	[31]
6-OHDA (substantia nigra)	rat	cystometry	adenosine A2A receptor antagonist and D1 and D2 receptor agonist	inhibition of adenosine A2A receptors suppresses the micturition reflex, the inhibitory effect of an adenosine A2A receptor antagonist is notaffected by D2 dopamine receptor stimulation and the inhibitory effects of D1 dopamine receptoractivation on bladder activity is prevented by priorapplication of an adenosine A2A receptor antagonist	[30]
**Genetic Models**					
**Model**	**Animal**	**Assessing Lower Urinary Tract Function**	**Intervention**	**Results**	**Ref**
α-synuclein overexpressing transgenic mice	mouse	size of urinary bladder		urinary bladder enlargement	[32]
α-synuclein overexpressing transgenic mice	mouse	conscious cystometry		Voided volume: smaller, Intercontraction intervals: reduced, nonvoiding contractions: increase	[14]
		urination patterns on filter paper		Urine spots: increase	
		NGF bladder content		increase	
GM2 Synthase knockout mice	mouse	urination patterns on filter paper		small void spots were significantly increased in male and female mice (the timing of dysfunction period is difference in sex)	[15]
		bladder volumes measured by ultrasonography		urinary bladder enlargement	
GM2 +/−	mouse	urination patterns on filter paper	FTY720/fingolimod	FTY720 treated GM2 +/− mice have significantly fewer void spots and have more large void spots	[33]

**Table 2 ijms-21-06520-t002:** Effects of each of the dopaminergic agonist/antagonist administration routes.

**Intact Rat**				
**Type**	**Drugs**	**Route**	**Dose**	**Effects**
D1 agonist	SKF 38393	iv	0.01–3.0 mg/kg	n.c.
		icv	5 μg (2 μL, 3 min)	volume↑
		it	5 μg (6 μL)	n.c.
D1 antagonist	SCH 23390	iv	0.1–3.0 mg/kg	volume↓
D2 agonist	quinpirole	iv	0.01–0.4 mg/kg	volume↓
		icv	2 μg (2 μL, 3 min)	volume↓
		it	1 μg (6 μL)	volume↓↓
D2 antagonist	remoxipride	iv	0.1–1.0 mg/kg	n.c.
**Parkinson’s Disease Model Rat**				
**Type**	**Drugs**	**Route**	**Dose**	**Effects**
D1 agonist	SKF 38393	iv	0.5–1.0 mg/kg	volume↑
		icv	2 μg (2 μL, 3 min)	volume↑
		it	5 μg (6 μL)	n.c.
D2 agonist	quinpirole	iv	0.2–0.4 mg/kg	volume↓
		icv	2 μg (2 μL, 3 min)	volume↓
		it	1 μg (6 μL)	volume↓↓

SKF38393 (i.v.) significantly increased bladder capacity (BC) in 6-OHDA rats without any apparent effects in sham rats. SKF38393 applied intracerebroventricularly (i.c.v.) under urethane anesthesia also increased BC in 6-OHDA-lesioned rats and by a smaller increment in sham rats. In contrast, quinpirole (i.v.) significantly reduced BC in sham and 6-OHDA-lesioned rats. Intrathecal injection of quinpirole similarly reduced BC in sham and 6-OHDA-lesioned rats.↑; significantly increased ↓; significantly reduced, iv; intravenous administration, icv; intracerebroventricular administration, it; intrathecal administration, n.c.; no change.

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
