# Peer review of "Animal Model for Lower Urinary Tract Dysfunction in Parkinson’s Disease"

_ijms, 2020, doi:10.3390/ijms21186520_

Round 1
Reviewer 1 Report
This manuscript reviewed recently published research animal models for Parkinson's disease (PD) and the research projects regarding the disease affecting lower urinary tract functions.
The review is comprehensive and the contents are well described. However, in this review I did not find the researched regarding the urethral sphincter discoordination in the research model. In addition to detrusor overactivity, urethral sphincter dysfunction is frequently encountered in patients with PD and is difficult to treat. The authors may add the review of the pathophysiology of this topic to make this review more comprehensive.
Author Response
Thank you for your comment. The influence of higher centers on micturition is very complex. As Reviewer 1 described that urethral sphincter discoordination is huge problem for treating the lower urinary tract dysfunction in the patient with PD. In recent reports, detrusor sphincter dyssynergia (DSD) is a rare cause of voiding dysfunction in Parkinson’s disease. DSD was observed on voiding at a rate of 0–3% [Araki I, Kitahara M, Oida T et al (2000) J Urol 164:1640–1643][Berger Y, Blaivas JG, de la Rocha ER et al (1987) J Urol 138(4):836–838][Sakakibara R, Hattori T, Uchiyama T et al (2001) J Neurol Neurosurg Psychiatry 71:600–606]. And in previous reports, DSD developed in animals with complete suprasacral spinal cord injury. However, for now, we cannot use DSD model in animal mode of PD.
We added this pathophysiology of urethral function in revised manuscript.
P2, L61
And detrusor sphincter dyssynergia (DSD) is huge problem for treating the lower urinary tract dysfunction in the patient with PD. In recent reports, DSD is much less common condition in Parkinson’s disease. DSD was observed on voiding at a rate of 0–3%.
And P3, L127
And DSD developed in animals with complete suprasacral spinal cord injury model. However, for now, we cannot use DSD model in animal mode of PD.
Reviewer 2 Report
Dear authors!
well done paper!
some minor comments:
- Please introduce a section: "take home message": include in this section a short summary of the results of the review
- Please discuss the limitation of animal models in a separate section at the end of you manuscript (ethical Problems, only preclinical Evaluation in anamilas, in some cases equal to human conditions but not human!, animal models represent NOT rewal lif, not a real disease, so conclusion in assisation to humans are possible with some limitations, because they cannot replace studis on human beeings.
I recommend publication, after adaptation according my comments...
Author Response
Thank you for your comment.
- We added the section “take home message” P13, L200 section 9
We reviewed various type of animal model for lower urinary tract dysfunction in PD. Animal research efforts have been primarily focused on PD motor signs and symptoms. However, almost all model has also non-motor dysfunction in these models. To detecting lower urinary tract dysfunction, many researchers focused on both bladder function and urethral function and, refined procedures were reported. These techniques were established and comparable to clinical human study. At present time, PD experimental models can be roughly classified into two categories: toxin-based and genetic models. All have their distinctive characteristics, so we have to fully understand each model. And many researchers in the PD field prefer the use of rodent animal models, before confirming concept and treatment effectiveness in a large animal study. Recently, some groups revealed new treatment option for lower urinary tract dysfunction in PD using animal models. The use of animal model studies is essential for understanding lower urinary tract dysfunction and for testing the efficacy and safety of promising treatment options.
- We discussed the limitation of animal models in a separate section at the end of this review (P12L176 section 7).
We reviewed animal model for lower urinary tract dysfunction in PD. However, to date, no ideal animal model has been agreed upon for PD research. Furthermore, it is difficult to develop a model that can fully recapitulate the features of human PD, which usually take years to manifest. Pre-clinical studies are vital to ensure safety and efficacy of new treatments before widely used in clinical practice, however, there are significant differences in the anatomy and biomechanics of different animal models and of humans. Obviously, animal study must be refined or altered in any way possible so as to decrease potential for suffering for all involved animals. Despite these limitations, the current animal models can provide a useful platform for selectively studying the pathophysiology and the interactions of the multiple etiologic factors involved in PD.